# Nutritional Approach Targeting Gut Microbiota in NAFLD—To Date

**DOI:** 10.3390/ijerph18041616

**Published:** 2021-02-08

**Authors:** Małgorzata Moszak, Monika Szulińska, Marta Walczak-Gałęzewska, Paweł Bogdański

**Affiliations:** 1Department of Treatment of Obesity, Metabolic Disorders and Clinical Dietetics, Poznan University of Medical Sciences, 61-701 Poznań, Poland; monikaszulinska@ump.edu.pl (M.S.); pbogdanski@ump.edu.pl (P.B.); 2Department of Internal Medicine, Metabolic Disorders, and Hypertension, Poznań University of Medical Sciences, 61-701 Poznań, Poland; mwalczakgalezewska@ump.edu.pl

**Keywords:** gut microbiota, diet, NAFLD, NASH, probiotics, prebiotics, bioactive substances

## Abstract

Non-alcoholic fatty liver disease (NAFLD) is a significant clinical and epidemiological problem that affects around 25% of the adult global population. A large body of clinical evidence highlights that NAFLD is associated with increased liver-related morbidity and mortality and an increased risk of cardiovascular disease, extrahepatic cancers, type 2 diabetes, and chronic kidney disease. Recently, a series of studies revealed the pivotal role of gut microbiota (GM) dysbiosis in NAFLD’s pathogenesis. The GM plays an essential role in different metabolic pathways, including the fermentation of diet polysaccharides, energy harvest, choline regulation, and bile acid metabolism. One of the most critical factors in GM stabilization is the diet; therefore, nutritional therapyappearsto be a promising tool in NAFLD therapy. This paper aims to review the current knowledge regardingthe nutritional approach and its implications with GM and NAFLD treatment. We discuss the positive impact of probiotics, prebiotics, and symbiotics in a reverse dysbiosis state in NAFLD and show the potential beneficial effects of bioactive substances from the diet. The full description of the mechanism of action and comprehensive examination of the impact of nutritional interventions on GM modulation may, in the future, be a simple but essential tool supporting NAFLD therapy.

## 1. Introduction

Non-alcoholic fatty liver disease (NAFLD) is a significant clinical and epidemiological problem [1]. Researchers have estimated that non-alcoholic fatty liver disease (NAFLD) affects around 25% of the adult global population [1,2]. Excessive lipid accumulation in hepatocytes is a consequence of an imbalance between the number of lipids delivered to the liver or formed de novo and their secretion in lipoproteins [3]. The disorder most often occurs with obesity, insulin resistance, type 2 diabetes (T2DM), dyslipidemia, hypertension, and metabolic syndrome [4].

Typically, the natural history of NAFLD starts with simple liver steatosis (non-alcoholic fatty liver, (NAFL) about 80–85% of cases), which can progress to active inflammation (non-alcoholic steatohepatitis (NASH)), fibrosis, cirrhosis, and hepatocellular carcinoma (HCC) [5]. Previous clinical evidence highlighted that NAFLD is associated not only with increased liver-related morbidity and mortality due to HCC [6] but also with an increased risk of cardiovascular disease (the most common cause of death in NAFLD), extrahepatic cancers (especially colorectal cancers), T2DM, and chronic kidney disease [2].

The pathogenesis of NAFLD is multifactorialand includes both genetic and epigenetic factors [7,8]. Of the risk factors for NAFLD progression, two: diet and dysbiosis in the gut microbiota (GM), have received increasing attention. Numerous studies have described that diet in NAFLD patients is characterized by poor composition, the overconsumption of simple carbohydrates, fructose, total and saturated fats (especially from red meat), and insufficient omega-3 fatty acids and dietary fiber intake [9,10,11]. The relationship between energy consumption, dietary macronutrients/micronutrients, and NAFLD’s onset and progressionhave been well described [11]. However, the final impact of a diet on the human metabolic profile results from a long-term nutrition model and not from the consumption of a single dietary component.

Among the pathogenetic factors of NAFLD, GM’s disturbances are also mentioned [8]. Previous reviews highlighted the potential of therapeutic methods based on GM modulation as a promising approach in NAFLD patients [12]. Currently, there is no pharmacological agent approved to treat NASH. Lately the efficacies of several agents in NAFLD management, including an FXR agonist (obeticholic acid), a PPARα and PPARδ agonist (elafibrinor), a CCR2 and CCR5 antagonist (cencriviroc), or glucagon-like peptide-1 analogues (semaglutide) have been described [13,14].

Despite that fact, lifestyle interventions based on dietary restriction and physical activity remain the first-line treatment for NAFLD.

In the face of the growing number of scientific reports on the relationship between nutrition and GM, it seems reasonable to review the nutritional approach’s current knowledge and the implications with GM and NAFLD treatment. In this review, we discuss the positive impact of probiotics, prebiotics, and symbiotics in a reverse dysbiosis state in NAFLD and show the potential beneficial effects of bioactive substances from the diet. The growing number of scientific reports confirming the influence of probiotics, prebiotics and symbiotics in modulating GM, with the simultaneous lack of sufficient pharmacological treatments for NAFLD, make them currently regarded as a promising strategy in the treatment of fatty liver. A full description of the mechanism of action and comprehensive examination of the impact of nutritional interventions on GM modulation may, in the future, be a simple but essential tool supporting NAFLD therapy.

## 2. GM Dysbiosis and NAFLD

There has been a growing interest in the GM’s role and its implications for human health in recent years. The GM is defined as a multispecies community of microorganisms, including a wide variety of bacteria, fungi, viruses, and archaea, colonizing in the gut [15]. This “invisible organ” exertsa variety of effects on the host, including energy harvest, nutrient metabolism [16], immunity regulation [17], bile acid metabolism [18], and stabilizing the functions of the nervous system [19].

There is evidence that GM in patients with obesity, metabolic disorders, and liver fat accumulation is characterized by lower diversity and altered composition—a reduction in beneficial species and increase in pathogenetic microbiota [20]. Wong et al. [21] described that NASH patients have fecal dysbiosis with a lower abundance of *Faecalibacterium* and *Anaerosporobacter* and a higher abundance of *Parabacteroides* and *Allisonella*. The positive changes in GM—increase in *Bacteroidetes* andreduction in *Firmicute*s*—*correlate with improvement in hepatic steatosis [21]. The link between dysbiosis in the bacterial community and the bioactivity and severity of NAFLD was also described by Boursier et al. [22]. In their study, the *Bacteroides*and *Ruminococcus*abundance was significantly increased in NASH patients, and was independently associated with NASH–fibrosis patients [22].

The clinical evidence foraltered gut microbiota in NAFLD is summarized in the table below (Table 1).

Dysbiosis may affect liver function, promoting infection, inflammation, and insulin resistance. The relationship between GM deregulation and metabolic syndrome and the mechanism underlying the development of GM-induced metabolic disorders were described in our previous review [29]. AsGM plays an essential role in different metabolic pathways, including the fermentation of diet polysaccharides, energy harvest, regulation in choline, and the bile acid metabolism, dysbiosis is linked with NAFLD’s pathogenesis [30]. Several mechanisms contribute to liver dysfunction, in which the GM is directly involved.

GM’s essential functions influence the gut wall integrity, suppressing intestinal inflammation, and restoring the tight junction structure [31]. The anatomical and functional close relationship of the gut and liver causes dysbiosis alterations in the intestinal permeability, leading to endotoxemia and chronic inflammation [32]. As a result of the increased intestinal permeability, harmful components of Gram-negative bacteria, such as liposaccharides (LPS), peptidoglycans, orbacterial DNAmay flow to the liver via the portal circulationand induce broad deregulation of the metabolic pathways presented in the liver [33]. The role of LPS in obesity and metabolic disorders was corroborated in an animal study in which mice injected with LPS showed a similar phenotype (weight gain, insulinresistance, and NAFLD progression) tothose fed with a high-fat diet (HFD) [32].

Endotoxemia results in the activation of specific pathogen-recognizing receptors (TLR, toll-like receptors) on Kupffer cells, sinusoidal cells, biliary epithelial cells, and hepatocytes. Endotoxemia promotes the development of hepatitis by activating pro-inflammatory pathways [15]. In clinical practice, intestinal integrity and gut inflammation may be assessed using different biochemical markers, such as calprotectin or fatty acid-binding protein (FABP) [7,34,35]. Research revealed that NAFLD subjects had an increased level of calprotectin, which points to the possibility of the co-occurrence of enteritis in this group of patients [34].

Obesity is directly associated with an increased risk of NAFLD. The prevalence of NAFLD is 50–90% in obese subjects and correlates with the obesity rate. According to the latest data, hepatic steatosis occurs in 65% of subjects with grade I–II obesity (body mass index (BMI) = 30–39.9 kg/m^2^) and in 85% of patients with grade III obesity (BMI = 40–59 kg/m^2^) [36]. Both the excess body weight itself and an unhealthy diet (e.g., HFD) predisposed to overweight imply disturbances in the intestinal microbiota’s homeostasis.

The GM’s differences between obese and lean subjects have been well described in previous animal and human studies [37,38,39,40]. The pathologically altered GM in obesity is characterized by genes that participate in energy harvest and metabolism, especially genes promoting the digestionof complex plant polysaccharides to short-chain fatty acids (SCFAs) and the absorption of monosaccharidesfrom the gut lumen affecting de novohepatic lipogenesis [41]. The increased accumulation of fat in the liver is also linked with the influence of GM dysbiosis on reduced fasting-induced adipose factor (FIAF) secretions, leading to the activation of lipoprotein lipase (LPL) [42].

Intestinal dysbiosis also affects the choline metabolism and trimethylamine N-oxide (TMAO) production. Choline is a constituent of cell and mitochondrial membranes and aneurotransmitter [43]. It is primarily supplied to the body fromfood asonly a small amount of choline can be synthesized internally. Physiologically, dietary choline (by the metabolite—phosphatidylcholine) modulatesvery-low-density lipoprotein (VLDL) production in the liver and participates in bile homeostasis. There is a direct relationship between a methionine–choline deficient (MCD) diet and NAFLD induction, which has been noticed in many animal studies [44,45]. Choline depletion caused by GM destabilization affects disturbances in the cholesterol metabolism and fatty acid oxidation and contributes to elevated transformation of trimethylamine (TMA) to TMAO in the liver [46]. High conversion of choline to TMA may reduce its bioavailability, affectingVLDL formation with the increased accumulation of triglycerides in the liver, which may promote fatty liver.

High levels of TMAO also promoteatherosclerosis through mechanisms related to lipid metabolism and inflammation [47] and predisposition to T2DM [45]. Hepatic flavin monooxygenase 3 (FMO3), whose expression is regulated by bile acid-activated farnesoid X receptor (FXR) [48], participatesin the oxidation of TMA to TMAO. Due to its protective effect on the lipid and glucose metabolism and the modulation of endogenous bile acid levels, the same receptor has beneficial therapeutic effects in NASH. The strong bi-directional relationship between the GM and FXR-mediated bile acid metabolism is another factor linking intestinal dysbiosis with NAFLD progression and exacerbation [49].

Both bile acids, due to their bacteriostatic properties, can modulate the microbial intestinal community. The individual gut bacterial patternaffects the bile acid synthesis and conversion of primary bile acids into secondary bile acids [50]. Parseus et al. [51] described that the GM was implicated in the bile acid profiles and signaling through the influence of FXR. As described, the impaired FXR signaling pathway inducedhepatic de novo lipogenesis, affected VLDL and triglyceride turnover, and damaged fatty acid oxidation.

Another important mechanism contributing to NAFLD’s occurrence is the increased endogenous production of ethanol through an altered bacterial community. The previous animal and clinical studies reported elevated blood ethanol (non-dietary) concentration in NAFLD compared within a healthy control [23,52,53]. Research showed that, in patients with NASH, the number of *Proteobacteria* (mainly *Escherichia coli*), which is responsible for the production of alcohol, increased [23]. For example, 1 g of *Escherichia coli* produces 0.8 g of ethanol from the fermentation of carbohydrates per hour in an environment without oxygen availability. Increased endogenous alcohol synthesis contributes to the destruction of tight connections and increases intestinal permeability, which, in turn, initiates liver damage [21].

## 3. NADLF, GM, and Diet—The Multidirectional Link

The diet is an important modifiable factor influencing the health state of the liver and shaping the GM. This multidirectional, well-documented relationship prompts thesearch for nutritional strategies that will improve metabolic and liver-related markers in NAFLD by restoring the homeostasis of the intestinal microbiota. This section describes the current dietary recommendations in NAFLD, emphasizing their effects on the gut bacterial community.

The assessment of diet in patients with NAFLD demonstrated that, in comparison with the controls, NAFLD subjects had higher or similar daily total calorie intakes. Still, the composition of macronutrients was disturbed [54,55]. For example, research revealed that NAFLD patients, independently of age, BMI, and total calories, had a higher intake of soft drinks and meat and a lower intake of omega-3 polyunsaturated fatty acid (PUFA)-fish [9].

The strong relationship between higher simple carbohydrate consumption (particularly fructose from soft drinks) was also demonstrated by other researchers [56]. In an observational study, Musso et al. [57] noticed that NASH patients consumed more saturated fat and less PUFA than healthy controls. However, Wehmeyer et al. [55] did not confirm this observation. Generally, it can be concluded that the Western-style diet (characterized by the overconsumption of calories, fat, simple sugars, processed foods, and a deficient in dietary fiber and unsaturated fatty acids) is more likely to be at risk of developing NAFLD than other nutritional patterns.

As there is currently no pharmacotherapy approved in NAFLD, the existing recommendations for patients focus on lifestyle modification strategies. The main goal of diet and exercise is a 7–10% weight loss, which correlates with histological outcome improvement [58]. Studies recommend avoiding simple sugars (especially fructose) and processed red meat and to increase dietary fiber intake and unsaturated fatty acids. The daily diet should also include lipid-lowering, insulin sensitivity-improving, and anti-inflammatory bioactive substances, as proven in recent years [59,60].

### 3.1. Body Mass Reduction and Influence of Weight Loss on GM Modulation in NAFLD

The American Association for the Study of Liver Diseases (AASLD) NAFLD practice guidelines recommend at least 3 to 5% weight loss to improve hepatic steatosis and a 7–10% body mass reduction to enhance the histologic features of NASH. Similarly, the guidelines from the European Association for the Study of the Liver-European Association for the Study of Diabetes-European Association for the Study of Obesity (EASL-EASD-EASO), and Korean Association for the Study of the Liver (KASL) recommend a weight reduction of 7 to 10% for the improvement of NAFLD. Both the EASL-EASD-EASO and AASLD guidelines recommend a 500–1000 calorie restriction to induce a weight loss of 0.5 to 1.0 kg/week, while the KASL guidelines suggest lower diet energy reduction (400 to 500 kcal) [58,61,62].

In the last decade, many studies described the change in a microbial community affected by an abnormal excess of body weight and fat mass, characterized by increased *Firmicutes* and decreased *Bacteroidetes* with greater adiposity [37,40,57]. The obese GM phenotype is characterized by advanced energy harvest associated with the genes responsible for expressing enzymes that breakdown indigestible complex plant polysaccharides to produce SCFAs [41,63], which promote weight gain.

Additionally, the alteration of GM composition may stimulate obesity and metabolic disorders by several mechanisms involving metabolic endotoxemia, alterations in the bile acid metabolism, and the influence of microbial metabolites: SCFAs, TMAO, indoles, and LPS on various metabolic pathways [29]. As described in the previous section, the distinct composition and ratio of *Firmicutes, Bacteroidetes*, and *Actinobacteria* at the level of phylum, family, and genus compared with the healthy group have been found in obese children and adolescents with NAFLD/NASH [23]. The alteration in microbial ecology has consequences in the mechanism leading to hepatic steatosis, inflammation, and fibrosis. Therefore, the dietary strategy focused on mass body reduction via restoring a “healthy GM pattern” can be one of the treatment options in NAFLD.

However, there is no consensus regardingthe best macronutrient composition of diet leading to long-term body mass reduction andimprovement in hepatic factors in NAFLD. According to the EASL-EASD-EASO guidelines, low-to-moderate fat intake and a moderate-to-high carbohydrate intake or low-carbohydrate ketogenic diets or high-protein diets appearto be the best dietary strategy. Still, AASLD and KASL emphasized the need for further research concerning the macronutrient composition of the diet. KASL suggested the potential effectiveness of a non-carbohydrate/low-fructose diet and AASLD suggested the Mediterranean diet (MD) [58,61,62]. The meta-analysis described that dietary interventions based on a caloric deficit of 300–1000 kcal/d or specified daily calorie supply (1200–1800 kcal/d; 20–30 kcal/kg/d) typicallyled to a modest weight loss and improvement in alanine aminotransferase (ALT), aspartate aminotransferase (AST), alkaline phosphatase, gamma-glutamyl transferase (GGT), liver steatosis, and liver stiffness [64].

### 3.2. Simple Sugar (Fructose) Intake Reductionand Carbohydrate-Restricted Dietary Patterns

Independent of energy intake, the macronutrient composition of a diet is associated with NAFLD/NASH’sinduction and the progression of NAFLD/NASH [65]. Different epidemiological studies have described macronutrientswith a harmful or beneficial effect on the liver.

Previous long-term observational studies noticed that excessive simple carbohydrate consumption, especially fructose (mainly when administered in drinking water), is one of the main risk factors for developing fatty liver [66,67].

High carbohydrate intake is a significant stimulus of liver de novo lipogenesis and appears more likely to be associated with increased inflammation and NAFLD progression [68,69]. Fructose is a simple sugar naturally occurring in honey and fruits; however, in the last few decades, it has gained popularity in food production as an ingredient in sweeteners, including sucrose and high fructose corn syrup (HFCS).

In a randomized study, Maersk et al. [70] noticed that high sucrose-sweetened beverage intake increased fat storage in the liver, muscles, and visceral fat. The increased lipogenesis and lipid deposition was associated with upregulating the activity of sterol regulatory-element binding protein-1c (SREBP-1c) and carbohydrate-responsive element-binding protein (ChREBP) and promoting mitochondrial dysfunction as an effect of high fructose consumption [71]. The dietary source of fructose is also essential. Epidemiological studies have shown that fructose from fruit induces the fatty liver to a lesser extent than sweeteners because they contain constituents (vitamins, minerals, and antioxidants) that may combat the effects of fructose [72,73]. The overconsumption of fructose also inhibits fatty acids β-oxidation and, therefore, increasestriglyceride (TG) synthesis in the liver [74].

High administration of fructose also induces other metabolic disturbances (including insulin resistance, hypertension, and dyslipidemia) [75] and results in microbiota composition changes—the alteration in the GM affecting the gut permeability and increased endotoxin translocation [24]. Dietary copper–fructose interactions also determine the influence of fructose on GM. Astudy conducted by Song et al. [76] to assess the effect of different dietary doses of copper with/without high fructose on GM showed that both low- and high-copper diets led to liver injury associated with gut barrier dysfunction [76]. The increased gut permeability was associated with the increased abundance of *Firmicutes* and the depletion of *Akkermansia* [76]. The unique copper–fructose relationship in NAFLD progression results from the fact that high amounts of dietary fructose impair the intestinal copper absorption, contributing to enlarging liver damage [77].

Interestingly, one of the recent studies showed also, that the type of sweetener and its combination with an HFD selectively influenced the GM, endotoxemia, and bacterial gene enrichment ofmetabolic pathways involved in LPS and SCFA synthesis. In this study Sánchez-Tapia et al. [78] noticed, that sucralose and steviol glycosides intake were associated with the lowest GM α-diversity. Moreover, sucralose caused an increase in *B. fragilis* abundance, and in proinflammatory cytokines. Additionally, sucrose (especially in combination with HFD) leads to the highest metabolic endotoxemia, weight gain, and metabolic disturbances.

Carbohydrates (CHO) also include dietary fiber, which is inversely associated with the hepatic fat fraction and intrahepatic lipid [79]. In anopen-label, randomized controlled clinical trial that included 112 patients, 12 weeks of increased whole-grain consumption significantly decreased the NAFLD grades, as well as theserum concentration of ALT, AST, GGTP, and the systolic and diastolic blood pressure [80]. In NAFLD subjects under 6-months of an energy-restricted diet, increased consumption of insoluble fiber from fruit (≥7.5 g/day) was associated with improvements in the fatty liver index (FLI), hepatic steatosis index (HIS), and NAFLD liver fat score (NAFLD-LFS) [81]. Due to dietary fiber’s prebiotic properties, the recommendations for increasing the fiber consumption in NAFLD patients are strongly supported.

One of the significant nutritional recommendations for NAFLD subjects is to reduce the intake of simple carbohydrates, including fructose and HFCS. The study conducted onadolescent boys with NAFLD to compare the effect of an 8-week low-free-sugar diet (less than 3% of calories from sugar) vs. a typical diet showed remarkable improvement in hepatic steatosis and liver-related measurements in the interventional group [82]. However, it is not clear whether the positive effect of carbohydrate-restricted diets is mostly due to the restriction of the total carbohydrate supply orthesignificant reduction in the fructose supply.

Few studies directly examined the impact of low/restricted fructose diets on fatty liver markers; however, there are frequent uncontrolled studies. Volynets et al. [83] investigated the effects of a 6-month reduction in fructose intake (~50% of baseline) in 10 subjects with NAFLD, which resulted in a reduced hepatic lipid content, transaminase level, and anthropometric measurements but had no effect on the prevalence of bacterial overgrowth and the intestinal permeability. The 9-day restriction (4% of daily calorie intake) of fructose in children with a high overall fructose intake (>50 g/d) reduced the liver fat and de novolipogenesis compared to controls on an isocaloric diet [84].

The reduction in simple sugars intake can be achieved in various ways, e.g., through alow-carbohydrate diet, very-low-calorie diet, or ketogenic diet. A low-carbohydrate diet (LCD, 60–150 g CHO/d) is one of the most common diets for weight management and metabolic syndrome treatment. There are also diets with more restrictive CHO recommendations: a very-low-carbohydrate diet (<60 g CHO/d) or ketogenic diet (<50 g CHO/day), whosepopularity has grownin recent years. There areseveral studies that assessed the LCD in NAFLD management.

In 2003, Foster et al. [85] showed that, in comparison to low-calorie, high-carbohydrate, and low-fat diets, the low-carbohydrate, high-protein, and high-fat diet (Atkins diet) resulted in a greater initial (in first six months) weight loss and more significant improvement in certainrisk factors for coronary heart disease. However, in both diets, the subjects reported poor adherence and high attrition. In another study witha 3-month calorie-restricted diet (1100 kcal/d) with low-CHO (<50 g/d) or high-CHO (>180 g/d), no difference was observed in the decrease in intrahepatic fat between the groups [86].

Similarly, de Luis et al. [87] demonstrated an improvement in anthropometric measurements, blood pressure, homeostasis model assessment-insulin resistance (HOMA-IR), TG, low-density lipoprotein (LDL), total cholesterol (TC), and liver enzymes (ALT, AST, and GGT) in NAFLD patients under low-fat or low-CHO dietsregardless of which macronutrients they emphasized. The same conclusion was provided by Katsagoni et al. [88]. A meta-analysis, including 20 clinical trials, noticed that both low/moderate fat and moderate CHO diets had a similar positive effect on liver function.

With its drastic carbohydrate reduction, the ketogenic diet (KD) is one of the most popular diets for weight loss among patients but still causes concern for nutritionists and doctors. A comprehensive review of the literature published in 2020 by Watanabe et al. [89] confirmed these doubts and indicated important aspects of the KD that require further research. On the other hand, there are studies thatreported the beneficial influence of KD ketone bodies on inflammation and fibrosis in patients with NAFLD; however, it is unknown if the effects werecaused by a very low carbohydrate intake (and ketosis) or by natural calorie restriction during KD (independently from ketosis and the macronutrient ratio [90].

The pilot study of a low-carbohydrate ketogenic diet (<20 g/d of carbohydrate) on obesity-associated biopsy-proven fatty liver disease showed that 6months of KD was associated with weight loss [90]. In another study, concerned the “Spanish Ketogenic Mediterranean Diet”(SKMD)(unrestricted calorie intake, high omega-3 PUFA), significant improvements in the body mass, BMI, waist circumference (WC), LDL, TG, HDL, ALT, AST, fasting plasma glucose (FPG), and steatosis degree wereobserved [91]. Interesting results were provided inthe study by Ministrini et al. [92], in which morbidly obese patients (n-52) underwent a 25-day very-low-carbohydrate ketogenic diet (VLCKD). After the intervention, significant reductions in the body mass index, FPG, insulinemia, and lipid profile parameters wereobserved.

Additionally, VLCKD reduced the number of patients with severe liver steatosis and increased lysosomal acid lipase (LAL), which is involved in the pathogenesis of NAFLD. Concerning changes in the GM, studies confirmed the beneficial effects of KD. Xie et al. [93], in their interventional study, demonstrated that KD decreased the phylum *Proteobacteri*a (*Cronobacter*) and increased the phylum *Bacteroidetes* (*Prevotella, Bifidobacterium, andBacteroides*) in children with drug-resistant epilepsy.

Similarly, KD for six months caused an overall decrease in the mean species diversity, increased *Bacteroides,* and decreased *Firmicutes* and *Actinobacteria* in a study conducted among children with resistant epilepsy [94]. The positive effect of an isocaloric low-CHO diet with increased protein content on the liver fat and alteration in the GM in obese subjects with NAFLD was proven by Mardinoglu et al. [95]. After the intervention, they noticed decreases in the hepatic de novo lipogenesis, increased serum β-hydroxybutyrate concentrations, folate-producing Streptococcus, and serum folate concentrations, and downregulation of the fatty acid synthesis pathway and upregulation of folate-mediated one-carbon metabolism and fatty acid oxidation pathways.

Although the KD, and particularly the very-low-calorie KD, appearsto be a promising option to achieve significant weight loss and improvement in fatty liver markers or GM composition, there are still concerns about the adverse health consequences of this diet and the poor adherence. Therefore, a very-low-calorie diet or KD is not recommended in everyday clinical practice. However, the last meta-analysis included 20 studies published until May 2019. Theyconcluded that the overall drop out of VLCKD was 7.5%, similar to patients undergoing a standard low-calorie diet [96]. No adverse effects on the lipids profile or potassium level werefound. The same opinion about using a KD and its safety or contraindications was presented by Watanabe et al. [97]. Therefore, future guidelines should include a specific recommendation for this intervention in patients with metabolic syndrome and NAFLD.

### 3.3. Saturated Fat Reduction, and Increased MUFA and Omega-3 PUFA

The NAFLD development is influenced by the amount and type of fat normally consumed in the diet. From the nutritional value, dietary fats are divided into saturated (SFA) or unsaturated, including monounsaturated fatty acids (MUFAs) or polyunsaturated fatty acids (PUFAs). They affect human health differently, being harmful or beneficial. Epidemiological studies noted that, in NAFLD patients, the insufficient PUFAs intake and low dietary PUFAs/SFA ratio was related to liver steatosis [98,99].

SFA generally occurs in animal products, like red meat, cream, butter, and whole-milkdairy products. Some vegetable products are also available (coconut oil, palm oil, and palm kernel oil) and processed foods [65]. Their high consumption directly correlates with an increased risk of cardiovascular diseases and metabolic syndrome [100]. There is also a link between SFA intake and NAFLD progression via oxidative stress, insulin resistance, PNPLA3 expression in hepatocytes, intrahepatic fat storage, and the circulation ceramides level [101,102]. In contrast, a high intake of MUFA, which are naturally found in olive oil, avocados, and nuts, is inversely correlated with dyslipidemia, T2DM, cardiovascular disease (CVD), Alzheimer’s disease, cancer, and potentially with fatty liver [103,104].

Cortez-Pinto et al. [54], in 2006, described that NAFLD patients revealed a higher MUFAs consumption; however, another study [9] did not confirm this association. PUFAs include omega-3 PUFAs (especially eicosapentaenoic acid (EPA), docosahexaenoic acid (DHA), and omega-6 PUFAs (linoleic acid). The omega-3 to omega-6 ratio in the diet determineshumanhealth according to the assumption that the 1:1–2 ratio is associated with a low risk of civilization diseases. Research described that increased omega-6 PUFAs consumption is a risk factor for systemic inflammation, metabolic syndrome, and CVD [100]. In contrast, omega-3 PUFAs play an important role in anti-oxidative and anti-inflammatory defense and insulin sensitivity regulation.

The omega-3 PUFAs are important modulators of hepatic gene expression, influencing genes involved in the β-oxidation of lipids in the liver (peroxisome proliferator-activated receptor-γ (PPAR-γ))and genes involved in hepatic fatty acid synthesis and storage (SREBP-1 and ChREBP) [105]. The previous study noted that, in subjects with lower circulating levels of n-3 PUFAs, the fatty liver β-oxidation was reduced. Regarding the GM, researchers investigated a diet enriched in MUFA and PUFA and found increases in the *Bacteroidetes* to *Firmicutes* ratio and elevation ofthe amount of *Bifidobacteria* and *Akkermansia muciniphila*. In contrast, SFA stimulated the growth of *Bilophila* and *Faecalibacterium prausnitzii* and caused a reduction in the numbers of *Bifidobacterium, Bacteroidetes, Bacteroides, Prevotella, and Lactobacillus ssp*. [106,107].

The systematic review and meta-analysis (n-18) published by Musa-Veloso et al. [108] assessed that omega-3 PUFAs wereuseful in patients’ dietary management with NAFLD. They showed that supplementation with omega-3 PUFAs affected the improvements in 6 of 13 metabolic outcomes (TC, LDL, HDL, TG, HOMA-IR, and BMI) and ALT and GGT (but not in AST), in the liver content (MRI), and in the steatosis score (USG). The effectiveness of omega-3 PUFAs in NAFLD was also investigated by He et al. [109]. In their meta-analysis of seven randomized control trials (RCTs) involving 442 patients, the supplementation with omega-3 PUFAs beneficially influenced the ALT, TC, TG, and HDL and tendedto a lowering effect on the AST, GGT, and LDL.

The same positive result of omega-3 PUFAs administration (0.8–13.7 g/d; for 8–12 months) on decreased liver fat and aminotransferase level have been published in Parker et al.’s [110] meta-analysis (n-9). However, the results from the previous studies focusing on MUFA and PUFA supplementation in NAFLD are equivocal. Parker et al. [110] assessed the effect of 12 weeks of supplementation with omega-3 PUFA from fish oil (588 mg EPA + 412 mg DHA, combined with 200 mg antioxidant, coenzyme Q10/daily) or placebo (olive oil capsules) for 12 weeks in overweight men.

They revealed no significant effect for fish oil versus placebo for liver fat, liver enzymes, anthropometry, or body composition, including visceral fat. Arandomized, double-blind placebo-controlled trial using marine omega-3 PUFAs (4 g DHA + EPA; for 15–18 months) observed that high dose DHA + EPA treatment in 96 patients with NAFLD affected multiple pathways, involving blood coagulation, the immune/inflammatory response, and the cholesterol metabolism [111].

There was a positive effect of 12 weeks of virgin olive oil (the equivalent of 20% of their total daily energy requirement from olive oil) on ALT and AST, but not on the severity of steatosis in NAFLD patients undergoing a hypocaloric diet [112]. The study aimed to investigate the impacts of fish oil (good source of DHA and EPA) and perilla oil (good source of alpha-linolenic acid (ALA)) on GM in rats with HFD-induced NAFLD, and showed that both n-3 PUFA rich oils ameliorated HFD-induced hepatic steatosis and inflammation, modulated the abundance of Gram-positive bacteria in the gut, and positively influenced the number of *Prevotella* and *Escherichia*. This effect was slightly less noticed in the perilla oil group [113].

The disturbances in the GM are involved in the development of NAFLD, and the dietary fatty acid intake influenced liver function; therefore, the full understanding of the reciprocal interaction between GM and n-3/n-6 PUFA may be necessaryfor NAFLD treatment.

### 3.4. Proteins

Regarding nutritional recommendations, the requirement for protein and preferred dietary protein sources in NAFLD is marginalized. However, several studies demonstrated a relationship between high proteinintake, especially animal protein, and the risk of NAFLD [55,114,115].

An inverse association between vegetable protein and the FLI score was described by Rietman et al. [104]. Similarly, the positive effect of soy protein (via isoflavones) on insulin resistance and liver steatosis was observed by Yang et al. [116]. The relation of high animal protein intake and NAFLD progression resulted from a different mechanism. One of them elevated the amino-acid catabolism, which increased the liver lipid oxidation [117].

Another focus is on the harmful influence of meat (especially highlyprocessed) on lipids, theglucose homeostasis, and the CVD risk. The profile of protein consumption—animal-derived proteins vs. plant-derived—also determines the bacterial enterotypes in the gut. Previous studies showed that a high intake of animal-based protein was associated with increased bile-tolerant anaerobes, such as *Bacteroide*s, *Alistipes*, and *Bilophila* [118,119]. Additionally, a high-protein diet increased the count of *Streptococcus*, *E. coli/Shigella*, and *Enterococcus*, and decreased thebeneficial *Faecalibacteriumprausnitzii*and *Ruminococcus* [120].

Astudy compared different isocaloric diets: the free high-fat diet (FFAT), the restrictive high-fat diet (RFAT) group, the restrictive high-sugar diet (RSUG) group, and the high-protein diet (PRO), and showed that the FFAT group had higher body weight, visceral fat index, liver index, peripheral insulin resistance, portal LPS, serum ALT, serum AST, and liver TG compared with all other groups. In the PRO group, the increase in body weight and visceral fat and ALT, AST, LPS, fasting insulin, and HOMA-IR were the lowest compared to RFAT, RSUG, and FFAT, whichmay be associated with changes in the GM.

A 12-week high-protein diet demonstratedan increase in *Bacteroidetes, Prevotella, Oscillospira,* and *Sutterella bacteria*, and a decreased abundance of *Firmicutes*. In comparison, the FFAT group had an increased abundance of *Firmicutes, Roseburia,* and Oscillospira bacteria, an elevated Firmicutes to Bacteroidetes ratio and a decreased abundance of *Bacteroidetes, Bacteroides,* and *Parabacteroides* bacteria [121]. In contrast, a recently published study in patients with biopsy-proven NAFLD (n-61) showed that a higher daily intake of protein (18.0% vs. 15.8% of daily protein-based calories) was associated with worse histological disease activity (respectively, a NAFLD activity score of 5–8 vs. a NAFLD activity score of 0–4) and with a lower *Bacteroides* abundance and an altered abundance of several other bacterial taxa [122]. This negative correlation was dependent on the dietary serine, glycine, arginine, proline, phenylalanine, and methionine intake, whose primary food source is meat.

A small number of studies comprehensively assessed the impact of dietary protein quantity and quality on NAFLD-related outcomes, and the GM’s condition and suggestedavital need for further research in this area.

## 4. New Dietary Perspectives in NAFLD Treatment via GM Modulation

### 4.1. Probiotics as a Promising Approach Therapy in NAFLD/NASH

Probiotics are defined as “live micro-organisms which, when administered in adequate amounts, confer a health benefit on the host.” They must survive the transit to the gut, where they reverse an adverse GM to the healthy state [123]. Probiotics are a well-known and well-studied preparation used in clinical practice. Researchdemonstratedthat probiotic supplementation reversing intestinal dysbiosis positively affected theliver function parameters, improved the lipid and carbohydrate metabolism, and reduced the inflammation status [123]. Their potential for weight reduction and body composition has also been speculated [4,124]. Multiple experimental trials have shown the therapeutic effects of probiotics in animal models of NAFLD.

Liang et al. [125] described that compound probiotics (0.6 g × kg^−1^ × d^−1^ compound probiotics) reducedthe weight (visceral and total fat), modulated the GM (increased TM7 phylum and decreased the *Verrucomicrobia* phylum), changed the level of SCFAs, and inhibited the lipid deposition and chronic metabolic inflammation in rats fed an HFD. The positive influence of probiotics (*Lactobacillus johnsonii* BS15, 2 × 10^7^ CFU/0.2 mL or 2 × 10^8^ CFU/0.2 mL)) on the GM pattern, endotoxemia, intestinal permeability, hepatic inflammation, and oxidative stress in mice under a HFD was also noted by Xin et al. [126].

Similarly, multispecies probiotic therapy’s positive influenceusing VSL#3 (Lactobacillus plantarum, Lactobacillus delbrueckii, Lactobacillus casei, Lactobacillus acidophilus, Bifidobacterium breve, Bifidobacterium longum, Bifidobacterium infantis, and Streptococcus thermophilus) on NAFLD progression was proven in several animal studies [127,128]. The positive findings from experimental studies led scientists to use probiotics in NAFLD/NASH patients. Previous studies showed that probiotic supplementation could improve liver steatosis and positively modulate the metabolic parameters that are typically disturbed in NAFLD [12,123,129]. Interventions to date have mainly involved multispecies prebiotics with/or without the addition of prebiotic substances (especially combinations of Lactobacillus, Bifidobacterium, and Streptococcus; VSL# 3) taken for 8–24 weeks [12,123].

Several previous studies showed significant improvements in the serum levels of AST and ALT in response to probiotic therapy: for example, a study conducted by Aller et al. [130] based on 3-month *Lactobacillus bulgaricus* and *Streptococcus thermophiles* intake anda study conducted by Wong et al. [131] in which patients with NASH were supplemented with a probiotic formula containing *Lactobacillus plantarum, Lactobacillus bulgaricus, Lactobacillusacidophilus, Lactobacillus rhamnosus,* and *Bifidobacterium bifidum* for six months. The positive influence of 1-month therapy with the combined probiotic formula (*Bifidobacterium Lactobacillus*, *Enterococcus*, *Bacillus subtilis,* and *Enterococcus*) on the metabolic parameters (ALT, AST, and lipid profile) and inhibition of serum tumor necrosis factor-α (TNFα), enhancing adiponectin in agroup of 200 patients with NADLF was described by Wang et al. [132].

The influence of probiotics on the histological indicators of liver damage (inflammation, steatosis, and fibrosis) was also described in multiple studies. Manzhalii et al. noticed that a 12-week low-calorie/low-fat diet with probiotic therapy (a cocktail containing a combination of *Lactobacilli,* Bifidobacteria, and *Streptococcus thermophiles*) reduced hepatic inflammation and liverstiffness (assessed by USG) and also positively modulated the GM [133]. A double-blind single-center RCT of live multi-strain probiotic (14 probiotic bacteria genera *Bifidobacterium*, *Lactobacillus*, *Lactococcus*, and *Propionibacterium*) in T2DM patients with NAFLD also provided evidence for the usefulness of probiotics in lowering the FLI and liver stiffness (LS) measured by shear wave elastography (SWE).

Additionally, 8-week therapy caused desirable ALT changesand AST, GGT, serum lipid, and cytokine (TNF-α, IL-1β, IL-6, IL-8, and IFN-γ) levels. The positive effect of probiotics (3 × 10 CFU/mL *Lactobacillus acidophilus*, 6 × 10 CFU/mL *Bifidobacterium lactis*, 2 × 10 CFU/mL *Bifidobacterium bifidum*, and 2 × 10 CFU/mL *Lactobacillus rhamnosus*) on sonographic and biochemical NAFLD in 64 obese children and adolescents was documented by Famouri et al. [134]. A similar, positive influence of VSL#3 on steatosis and liver enzymeswas noticed in another study [135].

Grąt et al. [136] demonstrated that the continuous administration of probiotics containing four strains: *Lactobacillus* Rosell^®^-1052, *Lactococcus casei* Rosell^®^-215, *Bifidobacterium bifidum* Rosell^®^-71, and *Lactobacillus helveticus* Rosell^®^-52 before liver transplantation effectively prevented postoperative infections and improvedthe early biochemical parameters of allograft function. As mentioned, four bacterial strains also havea brain–liver axis modulatory effect, antioxidative effect, and bile acid modulatory prosperities; therefore, they can potentially have a positive effect on NAFLD.

Probiotics positively influenced liver health and affected the different metabolic pathways. As CVD is the leading cause of death in NADFL, the positive effect of probiotics on vascular endothelial function is also essential. A previously published study concerning this relationship revealed that a multispecies probiotics supply positively modified both the functional and biochemical markers of vascular dysfunction [137].

The potential of using probiotic therapy to improve the biochemical indicators of fatty liver and glucose and lipid profiles has been the subject of several meta-analyses and systematic reviews. A meta-analysis, including four RCTs, on 134 NAFLD/NASH patients showed that probiotic therapy significantly improved the biochemical parameters of liver function (ALT and AST), reduced TC, high-density lipoprotein (HDL), HOMA-IR, and reduced the TNF-α level. However, probiotics therapy was not associated with BMI and glucose changes or the low-density lipoprotein (LDL) concentration [138].

Similarly, the analysis of nine RTC published before July 2015 with a total of 535 cases of NAFLD confirmed the influence of probiotics on ALT, AST, TC, HOMA-IR, and TNF-α but no improvement in the BMI, glucose, and insulin [139]. A meta-analysis published by Lavekar et al. [140] in the year 2017 showed that a variety of parameters, suchas BMI, AST, ALT, HOMA-IR, and the ultrasonic grade of liver steatosis, weresignificantly improved after probiotic treatment in different RCTs. One of the latest reviews in this area, conducted by Xiao et al. [129] in 2019, including 28 clinical trials (n-1555 NAFLD patients) published in April 2019, provided similar conclusions. The meta-analysis showed that 4–28 weeks probiotic therapy had beneficial effects on the ALT, AST, GGT, HOMA-IR, and BMI but not on the fasting blood sugar, lipid profiles, and TNF-α level.

Data from clinical trials conducted in the last two years on probiotic therapy’s effectivenessconcerning NAFLD confirmed the meta-analyses (Table 2).

### 4.2. Prebiotics and Synbiotics Modulate GM and Influence Steatosis in NAFLD/NASH

Prebiotics are defined as a “non-digestible food ingredient that induces particular changes in the composition and/or activity of the gastrointestinal microbiota and confers health benefits on the host.” They canselectively stimulate the growth and/or activity of one or a limited number of bacteria in the colonand therefore are essential growth agents for probiotics [123]. The representatives of prebiotics areoligosaccharides (such as fructooligosaccharides (FOS), galactooligosaccharides (GOS), isomaltooligosaccharides (IMO), xylooligosaccharides (XOS), lactulose, and soy oligosaccharides (SBOS)) and polysaccharides (e.g., inulin, cellulose, hemicellulose, pectins, and resistant starch).

All prebiotics have several properties: they (1) selectively stimulate the growth and activity of selected strains of bacteria with a beneficial effect on health, (2) lower the pH of the intestinal contents, (3) have beneficial local influence in the digestive tract, (4) are resistant to hydrolysis and different gastrointestinal enzymes, (5) are not absorbed in the upper gastrointestinal tract, (6) are a selective substrate for one or a certain number of useful species microorganisms in the colon, and (7) are stable in food processing [150].

The fermentation of prebiotics by GM produces SCFAs (including lactic acid, butyric acid, acetic acid, and propionic acid), which have multiple effects on human health. Various bacterial strains can ferment selected species of prebiotics, e.g., *Actinobacteria*, *Bacteroidetes*, and *Firmicutes* can ferment FOS, GOS, and XOS, while *Bifidobacterium* spp. is responsible for starch and fructans fermentation [151]. The mechanism of action and clinical applications of prebioticshave been investigated in previous experimental and clinical studies [123]. Their results provided evidence for the possibility of using prebiotics in the treatment ofmetabolic syndrome [152,153], gastrointestinal disorders [154,155], neurological disorders [156], dermatological problems [157], and bone metabolism [158,159].

The number of studies concerning prebiotics interventions in humans with NAFLD is still limited compared to probiotics; however, their results are promising. The oral administration of β-glucan derived from *Aureobasidium pullulans* (AP-PG) was effective in preventing the development of NAFLD in high-fat diet (HFD)-fed mice [160]. After 16 weeks of AP-PG intake, reduced serum TC, TG, ALT, and inhibition in TG accumulation (via influence on cholesterol 7 alpha-hydroxylase (CYP7A1) expression) in the liver wereobtained.

Another studyshowed that beta-glucan (1.5 g) intake for 12 weeks reduced the BMI, AST, ALT, TC, and TG and improved liver functions in humans [161]. Oat beta-glucans (1.5 mg/kg mc/d) also prevented metabolic disturbances, hepatic steatosis, and inflammation in LPS-induced NASH [162]. An RCT conducted by Akbarzadeh et al. [163] showed that 10 g of psyllium (*Plantago ovata*) administration was associated with a reduction in the ALT and AST, waist circumference, and calorie intake in obese patients with NAFLD.

The previous studies demonstrated that the positive influence of prebiotics on the course and progression of NAFLD resulted not only from the improvement in the metabolic outcomes and modulation of the GM composition but also from their capacity to increase SCFA—especially butyrate—production and reduce the expression of genes involved in lipogenesis and fatty acid elongation/desaturation [164].

Synbiotics are the combined use of prebiotics and probiotics [165]. Cortez-Pinto et al. [114] observed in an animal with high-fat choline-deficient diet (HFCD)-induced NAFLD that 18-week symbiotic (the combination of 10^11^ CFU of *Lactobacillus paracasei*: 25%, *Lactobacillus plantarum*: 25%, *Leuconostoc mesenteroides:* 25%, and *Pediococcus pentosaceus*: 25% with bioactive fibers) supplementation significantly reduced fibrosis, decreased endotoxemia, and modulated the GM pattern. Elsamparast et al. [166] described that 28-week lifestyle modification combined with synbiotics (200 million of seven strains: *Lactobacillus casei, Lactobacillus rhamnosus, Streptococcus thermophilus, Bifidobacterium breve, Lactobacillus acidophilus, Bifidobacterium longum*, and *Lactobacillus bulgaricus* with prebiotic: fructooligosaccharide) improved the fibrosis score in transient elastography (TE) in NAFLD patients.

Improvement in the liver’s ultrasound image in NAFLD patients was also noted after 24-week administration of 300 g/daily of symbiotic yogurt with 10^8^ CFU/mL *Bifidobacterium animalis* and 1.5 g inulin [167]. In the double-blind, placebo-control clinical trial, Javadi et al. [168] described that 3-month administration of probiotics (2 × 10^7^ CFU/day of *Bifidobacterium longum* and *Lactobacillus acidophilus*) and/or prebiotics (10 g/day of inulin improved aminotransferase enzymes, and supplementation with probiotics or pro- and pre-biotics recovered the grade of fatty liver in 75 subjects with NAFLD.

Alves et al. [169] described the influence of probiotics, prebiotics, and synbiotics on liver histopathology in hypercholesterolemic rats. They noticed that prebiotics (3 mg/d FOS) and synbiotics (3 mg/d FOS + 10^9^ CFU of each probiotic strain: *Lactobacillus paracasei* Lpc-37^®^ SD 5275^®^, *Lactobacillus rhamnosus* HN001^®^ SD 5675^®^, *Lactobacillus acidophilus* NCFM^®^ SD 5221^®^, and *Bifidobacterium lactis* HN019^®^ SD 5674) supplementation improved hepatic alterations via mediated gene expression related to β-oxidation (PPAR-α and CPT-1) and lipogenesis (SREBP-1c, FAS, and ME).

The study conducted in NASH patients also appearedpromising. Malaguarnera et al. [170] provided evidence that *Bifidobacterium longum* with FOS administration combined with lifestyle modification reduced the TNF-α, CRP, serum AST levels, HOMA-IR, serum endotoxin, steatosis, and the NASH activity index more than diet and exercise alone. The beneficial effect of symbiotic supplementation (*Lactobacillus reuteri* with guar gum and inulin) on hepatic steatosis, weight loss, and BMI, but not intestinal permeability LPS levels, were described in NASH patients [171].

Several meta-analyses of the effectiveness of probiotic/symbiotic therapy in NAFLD have been published in the last three years. Loman et al. [172] in 2018 analyzed 25 RTCs (9 used prebiotics, 11 used probiotics, and 7 used symbiotics; n-1309 NAFLD patients) and concluded that microbial therapies significantly reduced the AST and ALT, but not CRP. They noticed that serum TC and LDL resultsweremixed among prebiotics, probiotics, and symbiotics.

Similarly, Khan et al. [173], based on 12 probiotics/symbiotics RCTs (n-748 NAFLD patients), described that probiotics/symbioticswereassociated with a significant improvement in the ALT, AST, and liver fibrosis score (assessed by transient elastography). Theynoticed positive changes in the CRP, LDL, TG, and TC only for symbiotics. The ALT improvement, AST associated with reduced hepatic steatosis, and liver stiffness after probiotics/symbiotics therapy was presented in the meta-analysis published by Sharpton et al. [174] and Liu et al. [175].

For all three: probiotics, prebiotics, and symbiotics, several studies are currently being conducted, the results of which may decide on the advisability and effectiveness of their use in NAFLD patients (Table 2).

### 4.3. Vitamin E and Vitamin D and NAFLD

One of the nutritional factors associated with NAFLD development is the diet’s low nutritional value, including a low intake of vitamins, minerals, and bioactive compounds, such as polyphenols. Of the many vitamins involved in NAFLD’s pathogenesis, such as vitamin A, B vitamins, and vitamin C, most reports focused on describing the relationship between vitamin E and vitamin D deficiency and liver damage induction. Asvitamin D plays a vital role in the lipid and glucose metabolism and protects against oxidative stress and inflammation, its deficit leads to NAFLD progression.

Vitamin D inhibits liver fibrosis via the influence of the transforming growth factor-beta (TGF-β) pathway, activation of some hepatic TLP receptors, and a positive effect on insulin resistance [176]. A study on 12 weeks of calcitriol supplementation (25 µg/d) in 73 patients with NAFLD on a hypocaloric diet (reduction of 500 kcal per day) showed that, despite similar anthropometric changes observed in both groups (interventional and placebo), only in the calcitriol group was there areduction in the TG and AST and an increase in the HDL. Compared to placebo, the decrease in ALT, insulin, and HOMA was significantly higher [177].

Similarly, in RCT conducted among 109 patients of NAFLD diagnosed by USG and liver enzymes, supplementation with vitamin D3 (50,000 IU/day; for 12 weeks) affected increased serum vitamin D, and a decrease in HOMA-IR accompanied this rise in the ALT, AST, and serum CRP and increase in serum adiponectin [178]. Vitamin D also influencedthe GM composition [179]. Asystematic review of in vivo trials (n-24) suggested that low vitamin D levels may be associated with an increase in Bacteroidetes. Still, the relation between vitamin D and GM and NAFLD requiresfurther research [180].

Vitamin E is the major lipid-soluble antioxidant found in the human body. Due to its biological function, supplementation with vitamin E seems promising in NAFLD treatment [176]. Sato et al. [181] in 2015 analyzed five RCTs published until March 2014 and noticed that vitamin E significantly reduced the AST by −19.43 U/L, ALT by −28.91 U/L, and alkaline phosphatase (ALP) by −10.39 U/L, and improved steatosis by −0.54 U/L, fibrosis by −0.30 U/L, inflammation by −0.20 U/L, and hepatocellular ballooning by −0.34 U/L compared with the control group.

The results from a recent meta-analysis (n-15) concerning the effects of vitamin E supplementation in biochemical and histological parameters in adult patients with NAFLD indicated that vitamin E could be a promising tool in NAFLD treatment [182]. Vadarlis et al. [182] showed that vitamin E reduced the values of aminotransferases (−7.37 IU/L, 95% CI: −10.11 to −4.64 for ALT and −5.71 IU/L, 95% CI: −9.49 to −1.93 for AST), LDL, FPG, and serum leptin, and improved the liver pathology in every individual histologic parameter, especially for NASH patients.

In another RCT that aimed to compare five different interventions against NAFLD, only lifestyle modification, metformin (500 mg/day), silymarin (140 mg/day), pioglitazone (15 mg/day), and vitamin E (400 IU/day) for three months produceda significant improvement in the anthropometric parameters (BMI and WC), and a slight reduction in the ALT and AST in all treatment groups [183]. However, the influence of vitamin E supplementation on pediatric NAFLDis still underexplored. Amanullah et al. [184], in their systematic review and meta-analysis, concluded that only one of the four RCTs conducted in children with NAFLD noticed significant improvements in liver functions affected by the use of vitamin E. The same conclusion about the usefulness of using vitamin E in the population of children with NAFLD was presented by Sarkhy et al. [185].

The influence of vitamin E on the gut microbial community is poorly studied. The study conducted by Choi et al. [186] indicated that low-level consumption of vitamin E increased spleen and body weight and negatively changed the *Firmicutes* to *Bacteroidetes* ratio. However, the effect of vitamin E supplementation on changes in the GM in NAFLD/NASH patients is still lacking, and this relationship should be explored in further intervention studies.

### 4.4. Polyphenols in NAFLD Management

Polyphenols are bioactive compounds withbeneficial systemic effectsasdescribed in numerous studies. Polyphenols are a group with high chemical diversity, including resveratrol, quercetin, anthocyanins, epigallocatechin gallate, soy isoflavones, and silymarin. They are known for their well-described anticoagulant, lipid-lowering, blood pressure-lowering, antioxidant, and anti-inflammatory properties. Their positive effects against different pathologies, e.g., dyslipidemia, insulin resistance, T2DM, and hypertension, have been widely described [187].

The effect of polyphenols on human health depends in part on the individual GM pattern. On the one hand, the microbiota’s composition is modulated by the polyphenolic compounds, and the piece of the GM determinesthe catabolism of the ingested polyphenols into metabolites that are better absorbed and more active than the native phenolic compounds [188]. In a systematic review, Nash et al. [189] includedseven trials that assessed the effect of dietary grape and red wine polyphenols on the GM in humans and confirmed that GM modulated the ingested polyphenols and that increasing numbers of polyphenolic metabolites positively modulated the gut microbial ecology. The effects of different plant and fruit polyphenols have been tested in NAFLD/NASH patients with promising results (Table 3).

One of the most interesting bioactive compounds in metabolic syndrome management is berberine (BBR). BBR is an isoquinoline alkaloid naturally occurring in various plants, including the *Berberidaceae, Ranunculaceae*, and *Papaveraceae families* [190]. Interventions using BBR conducted in recent years have shown its potential in reducing the risk of developing CVD and metabolic diseases [191]. BBR is ascribed numerous health benefits, including reducing the lipid and cholesterol levels and improving insulin sensitivity [192].

Amulticenter, randomized, double-blind, placebo-controlled trial, conducted in patients with newly diagnosed T2DM, suggested that a12-week intervention with probiotics-plus-BBR may be a safe and effective option for supporting the treatment of T2DM [193]. In this trial, probiotics plus- BBR supplementation caused more significant changes in HbA1c than in the placebo group or probiotics-alone group. Several preclinical studies have suggested that BBR had hepatoprotective properties from various chemical insults [194,195,196].

The positive effect of BBR on the development and progression of NAFLD is also due to its influence onthe inhibition glucogenesis and hepatic lipogenesis [197], regulation of the adenosine monophosphate-activated protein kinase (AMPK) pathway, improvement of mitochondrial function, reduction in proprotein convertase subtilisin/kexin 9 (PCSK9) expression, and DNA methylation [198]. The therapeutic effect of BBR results from its impact on the GM (improvement of the composition and diversity as well as a reduction in endotoxemia) and maintaining the tightness of enterocyte connections, which may be beneficial in patients with NAFLD [199].

Turmeric is a spice frequently used in Ayurvedic medicine, with aregulative effect on the GM that has been widely tested in recent years. Turmeric products, curcumin, demethoxycurcumin, and bisdemethoxycurcumin, have been recognized as safe by several committees, including the Food and Drug Administration (FDA). The health-promoting properties of curcumin as an antioxidant, analgesic, antiseptic, antispasmodic, anti-inflammatory, and anticarcinogen have been documented [200].

Despite the poor bioavailability of curcumin, a few recent meta-analyses described that curcumin/turmeric alleviated hepatic steatosis [201] and positivelyinfluencedliver-related outcomes, like ALT and AST (especially in doses higher than 1000 mg/day) [202,203]. Jalali et al. [204], in ameta-analysis (n-9 RCTs), aimed to investigate the effects of curcumin on NAFLD and concluded that curcumin-based interventions resulted in ALT, AST, TC, LDL, FBS, HOMA-IR, and serum insulin reduction, but had no effects on the TG, HDL, HbA1c, body weight, and BMI.

No significant effect of turmeric/curcumin on body weight and BMI in NAFLD patients was described by Jafarirad et al. [205]; however, the last dose–response meta-analysis [206] (n-8 RCTs) showed that curcumin might positively affect the visceral fat and abdominal obesity. The beneficial systemic/metabolic effects of orally administered curcumin are also related to their influence on the microbial community. Turmeric/curcumin may promote the intestinal mucosal mechanical barrier [207] by upregulating the tight junction protein (occludin) and reducing the levels of TNFαand LPS.

The oral administration of curcuminalsosignificantly modulates the GM by increasing the abundance of *Bifidobacteria*, *Lactobacilli,* and butyrate-producing bacteria and reducing *Prevotellaceae*, Coribacterales, *Enterobacteria*, and *Enterococci* [208,209]. In ahuman study, turmeric and curcumin dietary supplementation [210] caused significantly but highly personalized changes in the gut microbiome (increases in most *Clostridium* spp., *Bacteroides* spp., *Citrobacter* spp., *Cronobacter* spp., *Enterobacter* spp., *Enterococcus* spp., *Klebsiella* spp., *Parabacteroides* spp., and *Pseudomonas* spp., and reduced abundance of several *Blauti*a spp. and most *Ruminococcus* spp.) in healthy subjects.

The effect of curcumin on the GM modulation in NAFLD is poorly investigated. One of the studies described that, in the HFD-induced NAFLD rat model, curcumin (200 mg/kg/day, for four weeks) decreased thirty-six potentially harmful bacterial strains and disrupted the GM pattern towards that of lean rats fed a normal diet [211]. Additionally, the curcumin treatment positively influenced the intestinal barrier integrity and reduced endotoxemia [211].

Silymarin is a mixture of flavonoids from *Sylbummarianum*, also known as milk thistle extract. The hepatoprotective effect of silymarin has been investigated in various liver diseases. In an animal model study, silymarin supplementation alleviated HDF-induced NAFLD and MCD-induced NASH [212,213]. In clinical trials among NAFLD/NASH patients, the positive effect of silymarin administration on liver-related outcomes has been observed [214,215]. The potential mechanism of silymarin action includes its participation in the SIRT1/AMPK pathway, the activation of FXR signaling or AMPK phosphorylation, and the modulation of antioxidative status [216]. There is a lack of research assessing the effect of silymarin supplementation on the GM diversity and activity in NAFLD/NASH patients.

Other polyphenols whose contribution to the prevention and inhibition of NAFLD progression and GM’s influence are still under discussion include resveratrol and green tea polyphenols.

Resveratrol is a phenolic compound with well-documented lipid-lowering, insulin-sensitizing, antioxidant, and anti-inflammatory properties. The potential multidirectional action of resveratrol also includes its influence on sirtuin 1 (SIRT1) activation, the suppression of hepatic lipogenesis, and induction of white adipose browning (via AMPK), activation of fatty acid oxidation processes, and increasing the rate of thermogenesis [216]. Resveratrol affected the GM composition by increasing the *Bacteroidetes* to *Firmicutes* ratios, inhibiting *Enterococcus faecalis*, and increasing Lactobacillus’s growthof *Bifidobacterium* [217].

Resveratrol also plays a vital role in gut barrier integrity stabilization [218]. Despite this, the effect of resveratrol against NAFLD is still ambiguous [219,220]. The last meta-analysis that assessed the resveratrol supplementation in NAFLD patients (seven RCTs, n-302 patients with NAFLD) showed that resveratrol supplementation, irrespective of the dose (500–3000 mg/d) or duration of administration (56–180 days), had no significant effect in terms of reducing anthropometric parameters, the lipid profile, glucose metabolism, or arterial pressure. Only one significant but surprising change was found in this analysis—an increased ALT level after resveratrol administration [221].

Epigallocatechin-3-gallate (EGCG), the most abundant polyphenolic catechin in green tea, has been widely investigated for its health-promoting properties [222]. According to Mansour-Ghanaei et al. [223], green tea may be a safe alternative approach for NAFLD treatment due to its influence on ALT, AST, BMI, TC, and LDL reduction. In contrast, ameta-analysis published in 2020 (n-15 RCTs) to assess the effects of green tea or green tea catechin on liver enzymes in NAFLD and healthy subjects showed that green tea’s overall impact on liver enzymes was non-significant. However, the effect of green tea on liver enzymes was dependent on the individuals’ health status with a moderate beneficial effect in patients with NAFLD [224].

The influence of green tea catechin has also been investigated in recent years. In animal studies, green tea polyphenol administration was associated with the genomic alterations of the gut-microbiome [225], and changes in the relative abundance of *Bacteroidetes* and *Fusobacteria* and proportions of *Acidaminococcus, Anaerobiospirillum, Anaerovibrio, Bacteroides, Blautia, Catenibactetium, Citrobacter, Clostridium, Collinsella*, and *Escherichia* [226]. The effect of green tea liquid consumption (400 mL per day) on increased Firmicutes to Bacteroidetes ratio, elevated SCFAs producing bacteria species, and reduction in endotoxemia was also noticed in healthy volunteers (n-12) [227].

## 5. Conclusions

NAFLD is a liver disorder with a high risk of progression to HCC and no approved pharmacotherapy to date. Therefore, lifestyle interventions based on diet and physical activity are the first-line treatment for NAFLD. The current nutritional recommendations for NAFLD include energy restriction, limiting the consumption of simple sugars (especially fructose), and supplying optimal omega-3 PUFAs and dietary fiber. Due to the described strong relationship between the diet, disturbance of the GM, and the development and progression of NAFLD, nutritional therapies focusing on remodulating the bacterial ecosystem appearto be particularly promising.

Previous studies highlighted the potential of probiotics, prebiotics, and symbiotics in GM restoration andamong NAFLD patients. There is also growing scientific evidence for the useof natural substances, such as polyphenols, in the therapy of this disease via GM modulation. However, there is still a need for research to thoroughly describe the effects of probiotics and bioactive compounds of the diet on the restoration of the pathological gut microbial ecosystem and on the improvement in liver-related outcomes in NAFLD/NASH patients. The results will allow thedevelopmentof a broader and more effective dietary approach for NAFLD.

## Figures and Tables

**Table 1 ijerph-18-01616-t001:** The clinical evidence describinggut microbiota (GM) deregulation in non-alcoholic fatty liver disease (NAFLD).

Reference	Population	Methods of GM Determination	GM Shift in NAFLD
Zhu et al. [23], 2013	63 children and adolescents divided into three groups: healthy, obese, and NASH	16S ribosomal RNA pyrosequencing of stool samples	↑*Bacteroides* ↑*Proteobacteria*/ (especially *Escherichia*) ↓*Firmicutes* (especially *Lachanospiraceae* and *Ruminococcaceae*)
Mouzaki et al. [24], 2013	15 subjects: 11 with simple steatosis (SS), 22 with NASH, and 17 healthy control (HC)	quantitative real-time polymerase chain reaction	↓*Bacteroidetes* and ↑ higher fecal *C. coccoides* in NAFLD vs. SS and HC
Raman et al. [25], 2013	obese NAFLD patients (n-30) vs. healthy controls (n-30)	multitag pyrosequencing	over-representation of *Lactobacillus* species and selected members of phylum *Firmicutes (Lachnospiraceae*; genera, *Dorea, Robinsoniella*, and *Roseburia*) in NAFLD patients, under-representation of phylum *Firmicutes* (*Ruminococcaceae*; genus, *Oscillibacter*)
Michail et al. [26], 2015	obese children with NAFLD (n-13) or without NAFLD (n-11), or lean children (n-26)	16S rRNA gene microarray, shotgun sequencing in stool samples, mass spectroscopy for proteomics and NMR spectroscopy for metabolite analysis	NAFLD*: higher Prevotella* and significantly higher levels of ethanol-production bacteria
Boursier et al. [22], 2016	57 patients with biopsy-proven NAFLD	16S ribosomal RNA gene sequencing of stool samples	*↑ Bacteroides (Genus)* *↑ Ruminococcus (Genus)* *↓ Prevotella (Genus)*
Del Chierico et al. [27], 2017	61 pediatric patients with NAFL/HASH or obesity and 54 HC	16S ribosomal RNA and volatile organic compounds	*↑ Anaerococcus (Actinobacteria), Ruminococcus (Firmicutes), Peptoniphilus (Firmicutes), Dorea* *(Firmicutes), Bradyrhizobium (Proteobacteria),* and *Propionibacterium acnes (Actinobacteria)* *↓ Rikenellaceae (Bacteroidetes), Oscillospira (Firmicutes)*
Da Silva et al. [28], 2018	39 adults with biopsy-proven NAFLD: 15 SS, 24 NASH, 28 HC	quantitative PCR metabolites analysis in feces and serum	HASH, SS:*↑Lactobacillaceae (Family) and Lactobacillus (Genus)* *↓ Ruminococcus (Genus), Faecalibacterium prausnitzii (Species), and Coprococcus (Genus)*

HC, healthy control; NAFL, non-alcoholic fatty liver; NASH, non-alcoholic steatohepatitis; NMR, nuclear magnetic resonance; SS, simple steatosis.

**Table 2 ijerph-18-01616-t002:** Summary of the recent clinical studies concerning probiotics, prebiotics, or symbiotics in NAFLD.

Reference	Population	Study Type	Intervention	Influence on NAFLD
Probiotics				
Duseja et al. [141], 2019	liver biopsy-proven adult patients with NAFLD (n-39) divided into: probiotic group (PG; n-20) and control (HC; n-19)	randomized, double-blind placebo-controlled clinical	multi-strain probiotic supplementation (2x 112.5 billion: *Lactobacillus paracasei,* *Lactobacillus plantarum*, *Lactobacillus acidophilus* *Lactobacillus delbrueckii* subsp. *bulgaricus*, *Bifidobacterium longum*, *Bifidobacterium infantis*, *Bifidobacterium breve*, and *Streptococcus thermophiles)* and lifestyle modification (regular exercises, dietary restriction) for 1 year	histological improvement (in hepatocyte ballooning, lobular inflammation, in the NAFLD activity score (NAS)) improvement in ALT, leptin, tumor necrosis factor-α (TNF-α), and endotoxins in PG vs. HC no adverse events
Ahn et al. [142], 2019	obese magnetic resonance imaging (MRI)-proven NAFLD patients (n- 68) divided into: probiotic (PG) or placebo groups (HC)	randomized, double-blind placebo-controlled clinical	the probiotic mixture contained 10^9^ CFU/1.4 g of six bacterial species (*Lactobacillus acidophilus,* *L. rhamnosus, L. paracasei,* *Pediococcus pentosaceus, Bifidobacterium lactis*, and *B. breve*) for 12 weeks	reduction in body mass, total body fat in PG but not in C a comparable reduction in intrahepatic fat (IHF) and triglyceride (TG) level in both groups weight.
Cai et al. [143], 2020	140 NAFLD divided into: control (HC; n-70) or probiotic group (PG; n-70)	observational study	HC: diet and physical activity PG: like HC+ combined *Bifidobacterium, Lactobacillus, Enterococcus* given orally, 1 g/time, 2 times/d for 3 months	better improvement in ALT, AST, GGT, TC, TG, HOMA-IR, NAS, and fecal flora conditions in the PG vs. HC
Prebiotics				
Wang et al. [144], 2019	8-week-old C57BL/6J male mice (n-40) fed HFD for 2 months	animal model, experimental study	MDG-1 (β-D-fructan polysaccharide extracted from the roots of *Ophiopogon japonicas*) vs. control	MDG-1 markedly blocked weight gain and ameliorated lipid accumulation, liver damage, and macrovesicular steatosis. MDG-1 restored GM gut microbiota balance (relative increase abundance of beneficial bacteria, especially SCFAs-producing bacteria), ↑ acetic acid and valeric acid, ↑ expression of hepatic phosphorylation of adenosine monophosphate-activated protein kinase, accompanying by regulating hepatic adipogenesis and adipocyte differentiation,
Takai et al. [145], 2020	12 newborn C57BL/6J male mice with monosodium glutamate-induced obesity	experimental study	5% FOS via drinking water for 18 weeks	FOS: improved the liver pathology and ↑ mRNA expression levels of lipid metabolism enzymes, inhibited adipocyte enlargement and formation of crown-like structures, ↑ fecal concentrations of n-butyric acid, propionic acid, and acetic acid
Bomhof et al. [146], 2019	14 liver biopsy-proven NASH patients	a placebo-controlled, randomized clinical trial	oligofructose (8 g/day for 12 weeks followed by 16 g/day for 24 weeks) vs. isocaloric placebo for 9 months	oligofructose: improved liver steatosis and NAS (independent of weight loss) ↑ *Bifidobacterium* and ↓*Clostridium cluster XI* and *I* no adverse side effects
Symbiotics				
Yao et al. [147], 2019	50 C57BL/6 mice with a high-fat diet (HFD)	animal model, experimental study	*Lactobacillus paracasei* N1115 (2.2 × 10^9^ CFU/mL in normal saline, 0.5 mL/day; N1115) or FOS (4 g/kg/day) or N1115 + FOS or placebo for 16 weeks	N1115, FOS, and synbiotics alleviated HFD-induced hepatic steatosis and release of TNF-α, and slowed the progression of cirrhosis. ↓ TC, FBG, fasting insulin, ↓ LPS, Toll-like receptor 4, and nuclear factor-κB. improved the intestinal barrier functions and histologic integrity (restoration of the p38 MAPK pathway and increased expression of occludin-1 and claudin-1)
Scorletti et al. [148], 2020	104 patients with NAFLD: symbiotic group (n-55) vs. placebo (n-49)	randomized double-blind placebo-control phase 2 trial of	symbiotic agents (FOS: 4 g/twice per day + *Bifidobacterium animalis subspecies lactis BB-12*) vs. placebo for 14 months	no significant difference in liver fat reduction between groups, symbiotic: ↑*Bifidobacterium* and *Faecalibacterium* species, ↓*Oscillibacter* and *Alistipes* species (the changes were not associated with liver fat or markers of fibrosis)
Abhari et al. [149], 2020	43 NADLF patients	randomized, double-blind, placebo-controlled clinical trial	10^9^ spores of Bacillus coagulans (GBI-30) plus 0.4 g/d inulin (n-26) vs. placebo (n-27) For 12 weeks	ALT, GGT, hepatic steatosis decreased significantly more in the synbiotics group vs. synbiotics supplementation significantly reduced serum TNF-α and nuclear factor-κB activity

ALT, alanine transaminase; AST, aspartate aminotransferase; FOS, fructo-oligosaccharides; GGT, gamma-glutamyl transpeptidase; HC, healthy control; HDF, high-fat diet; HOMA-IR, homeostasis model assessment-insulin resistance; IHF, intrahepatic fat; MRI, magnetic resonance imaging; NAS, NAFLD activity score; PG, probiotic group; TC, total cholesterol; TG, triglycerides; TNF, tumor necrosis factor.

**Table 3 ijerph-18-01616-t003:** Summary of the recent clinical study concerning polyphenols and other bioactive substances on GM and liver-related outcomes in NAFLD/NASH patients.

Reference	Population	Study Type	Intervention	Influence on NAFLD	Influence on GM
Song et al. [228], 2016	male C57BL/6J mice fed on HFD (n-36)	red pitayabetacyanins (HPBN) (200 mg/kg) for 14 weeks	animal model study (AMS)	HPBN:↓HFD-induced body weight gain, visceral obesity, and improved hepatic steatosis, adipose hypertrophy, and insulin resistance	↓*Firmicutes*, ↑*Bacteroidetes,* ↑*Akkermansia*
Li et al. [229], 2018	4-week-old male mice on a C57BL/6J mice with HFD-induced NAFLD (n-22)	diammoniumglycyrrhizinate (DG)(licorice root extracts)	AMS	↓ body weight, liver steatosis, hepatic inflammation	↑richness of GM ↓*Firmicutes to Bacteroidetes* ratio and endotoxin-producing bacteria (*Desulfovibrio*) ↑*Proteobacteri*a and *Lactobacillus* ↑ SCFA-producing bacteria (*Ruminococcaceae*, *Lachnospiraceae*↓ intestinal low-grade inflammation
Li et al. [230], 2019	mice fed 30% fructose water (HF) to induce NAFLD	polyphenol-rich loquat fruit extract (LFP) for 8 weeks	AMS	NAFLD prevention bodyweight, disordered lipid metabolism, oxidative stress, inflammation	↓endotoxin level, intestinal barrier protection, stabilization *Firmicutes* to *Bacteroidetes* ratio,
Cao et al. [231], 2016	BALB/C mice with HFDdiet-induced NASH divided into control (C), berberine (BBR), and model group (M) (n-30)	Berberine treatment C–standard diet; M- HFD BBR–HFD + berberine (200 mg/kg/d) for 13 weeks	AMS	BBR: ↓ body weight, serum levels of lipids, glucose, insulin, HOMA-IR, serum transaminase activity, and NAS, ↓ expression of CD14, IL-1, IL-6, and TNF-α	BBR: restored the relative level of *Bifidobacteria* and the *Bacteroidetes/Firmicutes* ratio
Feng et al. [211], 2017	4-week-old Sprague-Dawley (SD) male rats (n-30) with HFD-induced NAFLD	curcumin (200 mg/kg/d) for 4 weeks	AMS	decreased body weight, ALT, AST, liver weights reduction in hepatic lipid contents	improved intestinal barrier integrity, ↓ LPS, TNFα) *↓Spirochaetae*, *Tenericutes, Elusimicrobia* ↑*Actinobacteria*
Elvira-Torales et al. [232], 2019	44 male adult Sprague-Dawley rats with HFD-induced NAFLD	NC (normal diet), NB (normal diet + 2.5% spinach), NA (normal diet + 5% spinach), HC (high-fat diet), HB (high-fat diet + 2.5% spinach) and HA (high-fat diet + 5% spinach) For 5 weeks	AMS	spinach partially ameliorated HFD-induced alterations (FBG, TC, LDL, and liver fat accumulation)	↑*Lactobacillus*
Ushiroda et al. [233], 2019	male C57BL/6N mice fed with HFD	Green tea polyphenol (epigallocatechin-3-gallate) at a concentration of 0.32% for 8 weeks.	AMS	inhibited the increases in weight, the area of fatty lesions, and the triglyceride content in the liver	EGCG: ↑*Adlercreutzia*, *Akkermansia*, *Allobaculum* ↓*Desulfovibrionaceae*
Wu et al. [234], 2018	C57BL/6N mice with HFD-induced NAFLD (n-20)	0.5–1% Loniceracaerulea L. Berry Polyphenolsvs. for 45 days	Experimental study	significantly decreased the levels of IL-2, IL-6, MCP-1, and TNF-α in serum, as well as en-dotoxin levels in both serum and liver	*Modulation Firmicutes* to *Bacteroidetes* ↑*Bacteroides, Parabacteroides*, ↑*Bacteroidales* and *Rikenellaceae*, ↓the abundance of six bacterial genera belonging to the phylum *Firmicutes*, including *Staphylococcus*, *Lactobacillus*, *Ruminococcus*, and *Oscillospira*
Li et al. [235], 2019	Four-week-old mice with HFD-induced NAFLD (n-40)	sinapinesupplementation (500 mg kg^−1^ rapeseed oil, ≥98% purity) for 12 weeks	Experimental study	sinapinereduced the body weight and decreased TG and LDL, suppressed the intestinal NF-κB and TNF-α expressions, and enhanced the adipose tissue IRS-1 expression in the HFD mice inhibit the expression of inflammatory factors	↓*Firmicutes* to *Bacteroidetes* and increased the abundance of probiotics, such as *Lactobacillaceae*, *Akkermansiaceae* and *Blautia*
Van Hul et al. [236], 2018	C57BL/6J mice fed a high-fat diet (HFD)(n-56)	extracts from cinnamon bark (CBE; 2g/kg/d) and grape pomace (GPE; 8.2g/kg/d) for 8 weeks	Animal model study	CBE and GPE ↓ fat mass gain and adipose tissue inflammation without reducing food intake ↓ liver steatosis and lower plasma non-esterified fatty acid levels. beneficial effect on glucose homeostasis, improved glucose tolerance, and a ↓HOMA-IR	more profound change for the GPE than for the CBE, CBE: ↓*Peptococcus*, were GPE: ↓*Desulfovibrio, Lactococcus*, and ↑*Allobaculum* and *Roseburia*

## Data Availability

Not applicable.

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
