# Peer review of "Nutritional Approach Targeting Gut Microbiota in NAFLD—To Date"

_ijerph, 2021, doi:10.3390/ijerph18041616_

Round 1
Reviewer 1 Report
The paper does not address any of the studies and evidence indicating that probiotics, prebiotics and synbiotics can aggravate NAFLD. This needs to be remedied. There is no doubt that prebiotics can also unfavorably alter the gut microbiome to promote NAFLD.
Line 35. “which smoothly progresses”. Please rephrase. Poor word choice.
Many sentences in the abstract use the wrong tense. For example, affected should be affects, discussed should be discuss. Etc.
Line 42. This sentence is unscientifically worded: “The numerous studies described that diet in NAFLD patients is characterized by poor composition, overconsumption of simple carbohydrates, fructose, total and saturated fat (especially from red meat), and insufficient dietary fiber intake omega-44 3 fatty acids.”.
Line 52. This sentence is factually incorrect (and contains a typo). Although, in recent years, the efficacies of several agents in NAFLD menagement, including an FXR agonist (obeticholic acid), a PPARα and PPARδ agonist (elafibrinor),a CCR2 and CCR5 antagonist (cencriviroc), or glucagon like peptide-1 analogues (semaglutide) have been proved.
Line 71. This sentence is unscientifically worded. “It is known that” should be replaces by “There is evidence that” or “Several studies have indicated that”
Line 76. Please refrain from using “proven”. Nothing is proven in science, certainly not in the gut microbiota field.
Author Response
Author Response to Reviewer 1 Comments
Dear Reviewer,
We are grateful to you for taking Your time to read our paper and for your constructive comments. We have carefully reviewed the comments and have revised the manuscript accordingly. Our responses are given below in a point-by-point manner. Changes to the text are highlighted in the attached document using the "Track Changes" function in Microsoft Word, and the sections added to the manuscript are marked in yellow.
We hope the revised version is now suitable for publication and look forward to hearing from you in due course.
Sincerely,
Małgorzata Moszak
...................................................................................................................
Response to Reviewer 1 Comments
Point 1. The paper does not address any of the studies and evidence indicating that probiotics, prebiotics and synbiotics can aggravate NAFLD. This needs to be remedied. There is no doubt that prebiotics can also unfavorably alter the gut microbiome to promote NAFLD.
Response 1. Thank you for Your insightful review. Based on our previous and current experience, we have not observed that prebiotics can unfavorably alter the gut microbiome. Due to it, in our review paper, we focused on positive mechanisms and evidence indicating that probiotics, prebiotics, and synbiotics can prevent NAFLD. Thanks to Your precious comment, we will try to prepare a new review paper that addresses the studies and evidence indicating the negative influence of probiotics, prebiotics, and synbiotics on NAFLD.
Point 2. Line 35. “which smoothly progresses”. Please rephrase. Poor word choice.
Response 2. The sentence has been rephrased as follow:
line 35-37: “Typically, the natural history of NAFLD starts with simple liver steatosis (non-alcoholic fatty liver, (NAFL) about 80–85% of cases), which can progresses to active inflammation (non-alcoholic steatohepatitis (NASH)), fibrosis, cirrhosis, and hepatocellular carcinoma (HCC) [5].”
Point 3. Many sentences in the abstract use the wrong tense. For example, affected should be affects, discussed should be discuss. Etc.
Response 3. Thank you very much for your comments regarding correctness and language diligence, which will allow us to improve our work in the future. At the same time, we'd like to assure you that our manuscript was edited with the help of MDPI English editing (certificate in attached file) after your suggestion.
Point 4. Line 42. This sentence is unscientifically worded: “The numerous studies described that diet in NAFLD patients is characterized by poor composition, overconsumption of simple carbohydrates, fructose, total and saturated fat (especially from red meat), and insufficient dietary fiber intake omega-44 3 fatty acids.”.
Response 4. line 44-47:The sentence has been rewritten: „Numerous studies described that diet in NAFLD patients is characterized by poor composition, the overconsumption of simple carbohydrates, fructose, total and saturated fats (especially from red meat), and insufficient omega-3 fatty acids and dietary fiber intake [9–11].”
Point 5. Line 52. This sentence is factually incorrect (and contains a typo). Although, in recent years, the efficacies of several agents in NAFLD management, including an FXR agonist (obeticholic acid), a PPARα and PPARδ agonist (elafibrinor), a CCR2 and CCR5 antagonist (cencriviroc), or glucagon like peptide-1 analogues (semaglutide) have been proved.
Response 5. Thank you for your comments. The sentence you pointed has been corrected, and the next one has been rewritten:
line 54-58: „Lately the efficacies of several agents in NAFLD management, including an FXR agonist (obeticholic acid), a PPARα and PPARδ agonist (elafibrinor), a CCR2 and CCR5 antagonist (cencriviroc), or glucagon-like peptide-1 analogs (semaglutide) have been described [13,14]. Despite that fact, lifestyle interventions based on dietary restriction and physical activity remain the first-line treatment for NAFLD.
Point 6. Line 71. This sentence is unscientifically worded. “It is known that” should be replaces by “There is evidence that” or “Several studies have indicated that”
Response 6. Line 77-80. The sentence has been rewritten: „There is evidence that GM in patients with obesity, metabolic disorders, and liver fat accumulation is characterized by lower diversity and altered composition—a reduction in beneficial species and increase in pathogenetic microbiota [20].”
Point 7. Line 76. Please refrain from using “proven”. Nothing is proven in science, certainly not in the gut microbiota field.
Response 7.Thank you for your suggestion. All sentences where we using "proven" have been rewritten” as follows: line 82-84: “The link between dysbiosis in the bacterial community and the bioactivity and severity of NAFLD was also described by Boursier et al. [22].”
line 230-234: “In a randomized study, Maersk et al. [70] noticed that high sucrose-sweetened beverage intake increased fat storage in the liver, muscles, and visceral fat. The increased lipogenesis and lipid deposition was associated with upregulating the activity of SREBP-1c (sterol regulatory-element binding protein-1c) and ChREBP (carbohydrate-responsive element-binding protein) and promoting mitochondrial dysfunction as an effect of high fructose consumption [71].”
line 435-437: “Research demonstrated that probiotic supplementation reversing intestinal dysbiosis positively affected the liver function parameters, improved the lipid and carbohydrate metabolism, and reduced the inflammation status [122].”

Reviewer 2 Report
This manuscript is well documented and described, I just have some observations:
- Lines 60-62 Although the review goal is to discusse the impact of probiotics, prebiotics, and symbiotics in reverse a dysbiosis state in NAFLD, in the introduction is not clear why is this the main objective. Explain briefly why probiotics, prebiotics, and symbiotics would be an alternative treatment to NAFLD.
- Line 81: Table 1 has a different format in page 3 and 2.
- Section 2: It is not clear enough how dysbiosis is linked with NAFLD pathogenesis.
- Although the nutritional factors and metabolic disorders are well described and related to NAFLD, the probiotic and prebiotic role is poor (mechanisms or clear association direct or indirect to NAFLD).
Author Response
Author Response to Reviewer 2 Comments
Dear Reviewer,
We are grateful to you for your time to read our paper and for your constructive comments. We have carefully reviewed the comments and have revised the manuscript accordingly. Our responses are given below in a point-by-point manner. Changes to the text are highlighted in the attached document using the "Track Changes" function in Microsoft Word, and the sections added to the manuscript are marked in yellow. We hope the revised version is now suitable for publication and look forward to hearing from you in due course.
Sincerely,
Małgorzata Moszak
Response to Reviewer2 Comments
Point 1.Lines 60-62 Although the review goal is to discusse the impact of probiotics, prebiotics, and symbiotics in reverse a dysbiosis state in NAFLD, in the introduction is not clear why is this the main objective. Explain briefly why probiotics, prebiotics, and symbiotic should be alternative treatment to NAFLD.
Response 1. Thank you very much for Your suggestion. The following sentences (marked in yellow) have been added to the manuscript.
line 64-73: “In the face of the growing number of scientific reports on the relationship between nutrition-GM, it seems reasonable to review the nutritional approach's current knowledge and the implications with GM and NAFLD treatment. In this review, we discuss the positive impact of probiotics, prebiotics, and symbiotics in a reverse dysbiosis state in NAFLD and show the potential beneficial effects of bioactive substances from the diet. The growing number of scientific reports confirming the influence of probiotics, prebiotics and symbiotics in modulating GM, with the simultaneous lack of sufficient pharmacological treatments for NAFLD, make them currently regarded as a promising strategy in the treatment of fatty liver.A full description of the mechanism of action and comprehensive examination of the impact of nutritional interventions on GM modulation may, in the future, be a simple but essential tool supporting NAFLD therapy.”
Point 2. Line 81: Table 1 has a different format on page 3 and 2.
Response 2. Table 1 - the format has been standardized.
Point 3.Section 2: It is not clear enough how dysbiosis is linked with NAFLD pathogenesis.
Response 3.
Linking dysbiosis with NAFLD pathogenesis has been under investigation for about ten years, only. In Table 1, we presented research results based on less than 250 patients.
We still Reed more clinical research for a full explanation of dysbiosis and NAFLD pathogenesis influence.
Point 4. Although the nutritional factors and metabolic disorders are well described and related to NAFLD, the probiotic and prebiotic role is poor (mechanisms or clear association direct or indirect to NAFLD).
Response 4.
Thank you for Your insight full review. The lack of satisfactory description and relations of NAFLD with probiotic and prebiotics caused by very few clinical research results so far.

Reviewer 3 Report
In this review, the authors discussed the positive impact of probiotics, prebiotics, and symbiotic in counteracting a dysbiosis state in NAFLD, as well as the beneficial effects of bioactive substances from the diet.
Despite their work in the writing and referring to several research studies, I still found some missing information within the manuscript (some references are missing and sometimes recent research studies were not cited).
Moreover, if possible, I would suggest authors to add a picture embracing the main findings in the context of NAFLD and pro/prebiotics as well as the bioactive substances from the diet. This will help readers to have a visual view of what is known so far in this field.
I also add some other comments/remarks, please see below:
Line 54: a space is missing between the comma and the a
Line 60: Replace p with r (for review)
Line 76: And reduction in Firmicutes
Line 80: Biota has to be replaced with microbiota
Lines 90-91: Reference 31 is not the right one. There are other reviews more focused on the gut barrier function.
Lines 93-94: The LPS cannot be defined as a bacterial metabolites, but is a bacterial component of Gram negative bacteria. Flow to the liver via the portal circulation…
Lines 214-216: This is true mainly when fructose is administered in drinking water (see the literature, there are several animal studies).
Lines 234-238: High fructose consumption is also associated with a boost in Enterobacteriaceae (see article Suriano et al., Journal of Functional Foods, 2017). In addition, a recent study in rodents has shown that among the different type of sugars, the sucrose is the one mostly associated with higher metabolic endotoxemia, obesity and its related metabolic diseases (see Torres et al, Gut microbes, 2020).
Among the probiotics authors are no further describing the important role exerted by next-generation probiotics (i.e., Akkermansia muciniphila) in the context of liver diseases
Lines 491-492: I would more introduce that the concept of Prebiotics changed over the years, and now it is defined as “Non digestible….. Please refer to the latest review embracing how the definition of prebiotic change over the time.
Lines 504-505: Acetic acid is missing among the SCFAs
Line 507: Bifidobacterium sp, a p is missing. It has to be written has Bifidobacterium spp
Line 509: Did authors mean prebioics?
Line 545: A space is miising between 108 and CFU
Line 594: In vivo goes in Italique
Line 611: Typo error: Did authors mean NAFLD?
Line 626: A reference is missing, and the two brackets have to be deleted.
Line 625-627: Please rephrase this part a connection is missing between the two sentences.
Lines 635-636: Same comment as the line 626
Line 639: A space is missing between the reference and the test.
Line 645: Two times significant, please replace it with another world
Line 661: Lifer has to be replaced with life
Line 671: Occluding has to be replaced with Occludin
Line 671: TNFa and LPS
Line 697: A space is missing between browning and the bracket,
Line 698: A bracket is missing.
Lines 713: Nonsignificant, a space is missing
Line 721: SCFA, a s is missing, sometimes authors referred to SCFAs and sometimes to SCFA
Author Response
Author Response to Reviewer 3 Comments
DearReviewer,
We are grateful to you for Your time to read our paper and for your constructive comments. We have carefully reviewed the comments and have revised the manuscript accordingly. Ourresponsesaregivenbelow in a point-by-point manner. Changes to the text are highlighted in the attached document using the "track changes" function in Microsoft Word and the sections added to the manuscript are marked in yellow.
We hope the revised version is now suitable for publication and look forward to hearing from you in due course.
Sincerely,
Małgorzata Moszak
...................................................................................................................
Response to Reviewer3Comments
Point 1. In this review, the authors discussed the positive impact of probiotics, prebiotics, and symbiotic in counteracting a dysbiosis state in NAFLD, as well as the beneficial effects of bioactive substances from the diet.
Despite their work in the writing and referring to several research studies, I still found some missing information within the manuscript (some references are missing and sometimes recent research studies were not cited).
Moreover, if possible, I would suggest authors add a picture embracing the main findings in the context of NAFLD and pro/prebiotics as well as the bioactive substances from the diet. Thiswillhelpreaders to have a visual view of what is known so far in this field.
Response1. Thank you very much for your suggestion. We decided to add a graphical abstract (Figure 1) to our manuscript.
Point 2.Line 54: space is missing between the comma and the a
Response point 2-5, 10-28.Thank you very much for your comments regarding the correctness, language diligence, and numerous typos in our manuscript. At the same time, we'd like to assure you that our manuscript was edited with the help of MDPI English editing (certificate in attached file) after your suggestion.
line 54: space has been added.
Point 3. Line 60: Replace p with r (for review)
Response3.line 66. a typo has been corrected.
Point 4. Line 76: And the reduction in Firmicutes
Response4. line 84-85: The sentence has been corrected. “The positive changes in GM—increased in Bacteroidetes and reduction in Firmicutes—correlate with improvement in hepatic steatosis[21].”
Point 5. Line 80: Biota has to be replaced with microbiota
Response5. line 90: The word “biota” have been replaced with “microbiota”.
Point 6. Lines 90-91: Reference 31 is not the right one. Thereareotherreviewsmorefocused on the gut barrier function.
Response6.The reference has been replaced on:
- Chopyk, D.M.; Grakoui, A. Contribution of the Intestinal Microbiome and Gut Barrier to Hepatic Disorders. Gastroenterology. 2020, 159, 849-863. doi: 10.1053/j.gastro.2020.04.077
Point 7. Lines 93-94: The LPS cannot be defined as bacterial metabolites, but is a bacterial component of Gram negative bacteria. Flow to the liver via the portal circulation…
Response7.The sentence have been rewritten:
line 105-108: “As a result of the increased intestinal permeability, harmful component of Gram-negative bacteria, such as liposaccharides (LPS), peptidoglycans, or bacterial DNA may flow to the liver via the portal circulation and induce broad deregulation of the metabolic pathways presented in the liver [33].”
Point 8. Lines 214-216: Thisistruemainlywhenfructoseisadministered in drinking water (see the literature, there are several animal studies).
Response8.Your observation was added in the sentence:
line 228-230: “Previous long-term observational studies noticed that excessive simple carbohydrate consumption, especially fructose (mainly when administered in drinking water), is one of the main risk factors for developing fatty liver [66,67].”
Point 9.Lines 234-238: High fructose consumption is also associated with a boost in Enterobacteriaceae (seearticleSuriano et al., Journal of Functional Foods, 2017). In addition, a recent study in rodents has shown that among the different types of sugars, sucrose is the one most associated with higher metabolic endotoxemia, obesity and its related metabolicdiseases (see Torres et al, Gut microbes, 2020).
Among the probiotics, authors are no further describing the important role exerted by next-generation probiotics (i.e., Akkermansiamuciniphila) in the context of liver diseases
Response 9.Thank you for Your insight full review. Due to very little clinical research concerning the next generation probiotic, we have not presented the next generation probiotics in our paper. We run our probiotics researches and follow the results of other scientists. Thus it enables to prepare an upgrade of the present review.
Point 10. Lines 491-492: I would more introduce that the concept of Prebioticschangedover the years, and now it is defined as “Non digestible….. Please refer to the latest review embracing how the definition of prebiotic changes over time.
Response10.Thank you for Your comments. Further in lines 515-518 we present a novel definition of prebiotics, as „non-digestible…” We based on Perumpail et al. paper published in Diseases 2019.
Point 11. Lines 504-505: Aceticacidis missing among the SCFAs
Response 11. line 528-529 – missing acetic acid has been added.
Point 12. Line 507: Bifidobacteriumsp, a p is missing. It has to be writtenhasBifidobacteriumspp
Response12.line 531 – the typo has been corrected.
Point 13. Line 509: Didauthorsmeanprebioics?
Response13.line 532 – Thank you for your attention – We mean “prebiotics”. The mistake has been corrected.
Point 14. Line 545: Space is missing between 108 and CFU
Response14.line 572 – missing space have been added.
Point 15. Line 594: In vivo goes in Italique
Response15.line 594 – the typo has been corrected.
Point 16. Line 611: Typo error: Didauthorsmean NAFLD?
Response16.line 5642 – the typo has been corrected.
Point 17. Line 626: A reference is missing, and the two brackets have to be deleted.
Response17.The two brackets were unnecessarily placed there. The reference [186] concerned sentences in lines 653-658.
Point 18. Line 625-627: Pleaserephrasethis part a connection is missing between the two sentences.
Response18.Thank you for your comment, but we don't see the need to describe the effects of polyphenols in more detail. The indicated sentences line 653-658 are a short introduction to the discussion of research in the field of GM-polyphenols.
Point 19. Lines 635-636: Same comment as line 626
Response19.The two brackets were unnecessarily placed there.
Point 20. Line 639: Space is missing between the reference and the test.
Response20.line 670 – the missing space has been added.
Point 21. Line 645: Twotimessignificant, please replace it with another world
Response21.line 676-678. The sentence has been rewritten as follows: “In this trial, probiotics plus- BBR supplementation caused more significant changes in HbA1c than in the placebo group or probiotics-alone group.”
Point 22. Line 661: Liferhas to be replaced with life
Response22.line 694 – the mistakes in “liver-related” have been corrected.
Point 23. Line 671: Occludinghas to be replaced with Occludin
Response23.line 704: the mistakes have been corrected.
Point 24. Line 671: TNFa and LPS
Response24.line 704: the mistakes have been corrected.
Point 25. Line 697: Space is missing between browning and the bracket,
Response 25.line 732 - the missing space has been added.
Point 26. Line 698: A bracket is missing.
Response26.line 732 - the missing bracket has been added.
Point 27. Lines 713: Nonsignificant, space is missing
Response27.line 749 - the missing space has been added.
Point 28. Line 721: SCFA, as is missing, to SCFAs and sometimes to SCFA
Response28. line 757- the notation has been corrected.

Round 2
Reviewer 3 Report
Dear Authors,
Thanks a lot for all your changes done within the manuscript taking into account the reviewers' comments. However, I still found some important points that need to be adjusted in the review. Please see below my comments:
Lines 69-71: Figure 1: Please be aware of the image that you used. The one of the gut microbiota is commonly found on internet. I kindly ask the authors to check for the copyright.
Please replace metabolizm with metabolism
Line 91: Please check the table. Is there a missing reference for the decrease in Ruminococcus?
Lines 226-227: The references 66 and 67 are not the right ones. The first refers to a human study, the second one is a review. I kindly advice the authors to carefully check the references and to refer either to preclinical (rodents) or clinical studies (humans) (e.g., O’ Sullivan et al., 2014; Zhang et al., 2017; Suriano et al., 2018). Please refer to these studies and others when talking about fructose supplementation.
There is also a recent research study comparing the different types of sugars (Torres et al., 2020), and describing that the type of sweetener and an high-fat diet HFD modify the gut microbiota, bacterial gene enrichment of metabolic pathways involved in LPS and SCFA synthesis, and metabolic endotoxemia associated with different metabolic profiles.
Author Response
Dear Reviewer 3,
Thank you very much for your time to reread our manuscript and for your suggestions. Our responses are given below in a point-by-point manner. Changes to the text are highlighted in the attached document using the "Track Changes" function in Microsoft Word, and the sections added to the manuscript are marked in yellow. We hope the revised version is now suitable for publication and look forward to hearing from you in due course.
Point 1. Lines 69-71: Figure 1: Please be aware of the image that you used. The one of the gut microbiota is commonly found on internet. I kindly ask the authors to check for the copyright.
Please replace metabolizm with metabolizm
Response 1. In figure 1, we replaced the image using Servier Medical Art.
Point 2. Line 91: Please check the table. Is there a missing reference for the decrease in Ruminococcus?
Response 2.Thank you for your suggestion. We added a reference in line 91, and we checked the correctness of the table.
Point 3. Lines 226-227: The references 66 and 67 are not the right ones. The first refers to a human study, the second one is a review. I kindly advise the authors to carefully check the references and to refer either to preclinical (rodents) or clinical studies (humans) (e.g., O’ Sullivan et al., 2014; Zhang et al., 2017; Suriano et al., 2018). Please refer to these studies and others when talking about fructose supplementation.
Response 3. The references have been checked and corrected as follows:
Reference 66 – Suriano, F.; Neyrinck, A.; Verspreet, J.; Olivares, M. Particle size determines the anti-inflammatory effect of wheat bran in a model of fructose over-consumption: Implication of the gut microbiota. Journal of Functional Foods2018, 41, 155–162, doi:10.1016/j.jff.2017.12.035.
Reference 67 -O'Sullivan, T.; Oddy, W.; Bremner, A.; Sherriff, J.; Ayonrinde, O.; Olynyk, J.; Beilin, L. et al. Lower Fructose Intake May Help Protect Against Development of Nonalcoholic Fatty Liver in Adolescents With Obesity. Journal of Pediatric Gastroenterology and Nutrition 2014, 58, 624-631, doi:10.1097/MPG.0000000000000267.
Point 4. There is also a recent research study comparing the different types of sugars (Torres et al., 2020), and describing that the type of sweetener and an high-fat diet HFD modify the gut microbiota, bacterial gene enrichment of metabolic pathways involved in LPS and SCFA synthesis, and metabolic endotoxemia associated with different metabolic profiles.
Response 4. The following sentences including a recent study concerning different type of sugars and GM have been added to the manuscript:
Line 258-264:
Interestingly, one of the recent studies showed also, that the type of sweetener and its combination with an HFDselectively influenced the GM, endotoxemia, and bacterial gene enrichment of metabolic pathways involved in LPS and SCFA synthesis. In this study, Sánchez-Tapia et al. [78] noticed that sucralose and steviol glycosides intake were associated with the lowest GM α-diversity. What’s more, sucralose caused increase in B. fragilis abundance, and in proinflammatory cytokines. Additionally, sucrose (especially in combination with HFD) leads to the highest metabolic endotoxemia, weight gain, and metabolic disturbances.
